

# Herbivore corridors sustain genetic footprint in plant populations: a case for Spanish drove roads

Alfredo García-Fernández[1], Pablo Manzano[2,3,4], Javier Seoane[3], Francisco M. Azcárate[3], Jose M. Iriondo[1] and Begoña Peco[3]

[1] Área de Biodiversidad y Conservación, Universidad Rey Juan Carlos, Móstoles, Madrid, Spain
[2] Commission on Ecosystem Management, International Union for Conservation of Nature, Nairobi, Kenya
[3] Terrestrial Ecology Group—Departamento de Ecología, Centro de Investigación en Biodiversidad y Cambio Global (CIBC), Universidad Autónoma de Madrid, Madrid, Spain
[4] HELSUS, Faculty of Biological and Environmental Sciences, University of Helsinki, Helsinki, Finland

Corresponding author
Pablo Manzano,
pablo.manzano.baena@gmail.com

## ABSTRACT

Habitat fragmentation is one of the greatest threats to biodiversity conservation and ecosystem productivity mediated by direct human impact. Its consequences include genetic depauperation, comprising phenomena such as inbreeding depression or reduction in genetic diversity. While the capacity of wild and domestic herbivores to sustain long-distance seed dispersal has been proven, the impact of herbivore corridors in plant population genetics remains to be observed. We conducted this study in the Conquense Drove Road in Spain, where sustained use by livestock over centuries has involved transhumant herds passing twice a year en route to winter and summer pastures. We compared genetic diversity and inbreeding coefficients of *Plantago lagopus* populations along the drove road with populations in the surrounding agricultural matrix, at varying distances from human settlements. We observed significant differences in coefficients of inbreeding between the drove road and the agricultural matrix, as well as significant trends indicative of higher genetic diversity and population nestedness around human settlements. Trends for higher genetic diversity along drove roads may be present, although they were only marginally significant due to the available sample size. Our results illustrate a functional landscape with human settlements as dispersal hotspots, while the findings along the drove road confirm its role as a pollinator reservoir observed in other studies. Drove roads may possibly also function as linear structures that facilitate long-distance dispersal across the agricultural matrix, while local *P. lagopus* populations depend rather on short-distance seed dispersal. These results highlight the role of herbivore corridors for conserving the migration capacity of plants, and contribute towards understanding the role of seed dispersal and the spread of invasive species related to human activities.

## INTRODUCTION

Increased habitat fragmentation has been perceived as a major worldwide threat for ecosystems and biodiversity for the last decades (*Fahrig, 2003*). Fragmentation limits population size and reduces seed and pollen flow among populations, leading to genetic drift and reduction in within-population genetic diversity. This genetic impoverishment can limit evolutionary potential, promote selfing and therefore, decrease population viability through inbreeding depression. Such effects impact not only biodiversity conservation (*Ouborg, Vergeer & Mix, 2006*), but also ecosystem productivity (*Crutsinger et al., 2006*) and microevolutionary responses (*Matesanz et al., 2017*). Increasing connectivity between isolated vegetation patches is a major conservation strategy to reduce the negative effect of habitat fragmentation (*Hodgson et al., 2011*). Nevertheless, increasing effective dispersal of seeds or pollen among patches (functional connectivity) is not always possible by just increasing structural connectivity as, for example, by establishing landscape corridors, because the availability of dispersal vectors or recruitment microsites are needed to sustain functional connectivity (*Auffret et al., 2017*; *Plue, Aavik & Cousins, 2019*).

Dispersal is a central life-history trait that structures the properties of ecosystems (*Bonte & Dahirel, 2017*; *Massol et al., 2017*). Particularly among fragmented natural habitats, dispersal is a key element for reducing extinction risk, and is subjected to strong selective pressures (*Cheptou et al., 2017*). Indeed, forecasts for the ongoing climate change include a massive rearrangement of species distributions, which could track their climate niches only if their dispersal is not impeded. In this context, livestock-mediated seed dispersal is an important vector for increasing functional connectivity. While the ecological importance of large wild herbivores is practically disappearing at the global scale (*Ripple et al., 2014*; *Smith & Botha-Brink, 2014*; *Bar-On, Phillips & Milo, 2018*), as are the migratory systems they support (*Berger, 2004*), domestic herbivores have largely overtaken their role to the point that they can even substitute extinct megaherbivores (*Pires et al., 2014*). The effectiveness of livestock seed dispersal as a cheap and effective management tool for increasing connectivity between isolated grassland habitat patches has been extensively surveyed (*Cosyns et al., 2005*; *Couvreur et al., 2005*; *Auffret et al., 2012*; *Emmerson et al., 2012*), showing that the complementarity between different modes of dispersal (endo- and epizoochory) can lead to an almost complete representation of the grassland community in the dispersal spectra (*Couvreur et al., 2005*). Herbivory, trampling and nutrient redistribution by livestock also generate microsites favorable to the recruitment of many adapted species (*Bullock, Hill & Silvertown, 1994*; *Olff & Ritchie, 1998*), contributing to functional connectivity. Unfortunately, livestock management is changing in many regions of the world for economic and social reasons, and practices such as transhumance and rotational grazing are declining (*Poschlod et al., 1998*; *Olea & Mateo-Tomás, 2009*; *Auffret, Plue & Cousins, 2015*).

Drove roads—routes traditionally used for mobile pastoralist livestock in many regions of the world—become particularly interesting to promote functional connectivity between isolated grasslands. Historic ones are presumed to often derive from ancient migratory

routes of wildlife (*Manzano Baena & Casas, 2010*) Drove roads have been observed to have a strong effect in increasing multifunctionality of the landscape by working as linear grasslands and increasing heterogeneity at the large scale, translating into an increase of biodiversity in diverse taxonomic groups such as plants (*Azcárate et al., 2013a*), ants (*Azcárate et al., 2013b*; *Hevia et al., 2013*) or even bees, linked with important pollination services (*Hevia et al., 2016*). Given the proven capacity of livestock in achieving long-distance dispersal along drove roads, both by endozoochory (*Manzano, Malo & Peco, 2005*) and by epizoochory (*Manzano & Malo, 2006*), mobile pastoralism taking place in drove roads could potentially mitigate the consequences of population isolation, which is a major outcome of fragmentation processes (*Fahrig, 2003*; *Mitchell et al., 2015*), especially in a grassland context (*Pretelli, Isacch & Cardoni, 2018*).

Dispersal is a multi-faceted process where the quantification of every step implies complex measurements that can result in great uncertainties at the landscape scale (*Wang & Smith, 2002*), and estimating its effective footprint is much more feasible through genetic analyses. Also, the study of the genetic structure and the gene flow between populations are essential to assess their viability in the long term. Fragmentation by isolation leaves a measurable genetic trace (*Leblois, Estoup & Streiff, 2006*). With such background, studies in livestock-grazed rangelands have observed that grazed areas display a more homogeneous genetic structure than ungrazed ones (*Smith et al., 2009*) and a less-than-expected population differentiation in *Anthyllis vulneraria* fragments historically subjected to rotational grazing (*Honnay et al., 2006*). *Willerding & Poschlod (2002)* were unable to see effects of herd mobility in a rotational context, possibly due to the election of the target species (the anemophilous *Bromus erectus*). More recently, however, rotational grazing livestock has been observed to genetically link island grassland communities of *Campanula rotundifolia* that are otherwise isolated (*Plue, Aavik & Cousins, 2019*). Similarly, livestock herded by foot from patch to patch in a rotation scheme has shown to be strong enough to leave a measurable trace in plant populations according to the distance covered, reducing genetic structure and increasing the viability of re-introduced grassland plants for *Dianthus carthusianorum* (*Rico, Boehmer & Wagner, 2014a*; *Rico et al., 2014b*; *Rico & Wagner, 2016*) and *Pulsatilla vulgaris* (*DiLeo et al., 2017*). While these results highlight the role of livestock as an effective and determinant vector for gene flow across the landscape, the artificial, network-like management of rotational grazing differs from the more corridor-like type of dispersal that is expected from wild migratory herbivores (*Berger, 2004*), with bigger herds, longer distance migrated, shorter transit time per grassland area unit, and routes maintained during centuries, and which is mimicked by transhumant livestock (*Manzano Baena & Casas, 2010*). This perspective is even more relevant considering how widespread transhumance is worldwide, and how livestock can potentially substitute the ecological functions of wild migratory herbivores.

There are other important gene flow vectors beyond seed dispersal, however. Pollen flow also depends on landscape connectivity, where the structural features of the landscape that harbors pollinator communities (*Auffret et al., 2017*) are more relevant for it than the existence of functional seed-dispersal vectors. This is a factor that seems especially

relevant for fragmented agricultural landscapes crossed by livestock corridors, as the latter are reported to harbor a significant pollinator community (*Hevia et al., 2016*). Wind pollination in angiosperms seems to have evolved in cases where biotic pollinators are limited (*Culley, Weller & Sakai, 2002*) and long-distance flow would be achieved in less common cases of high wind turbulence and speed, both very dependent on topography (*Auffret et al., 2017*). Persistent seed banks can also alter the effects observed in the genetic structure of plant populations, particularly by delaying or mitigating the effects of fragmentation (*Honnay et al., 2008*; *Plue et al., 2017*).

In this study, we aimed to check whether dispersal processes in drove roads subjected to active use influence the population genetic structure of *Plantago lagopus*, an annual selfing plant species dispersed by livestock endo- and epizoochory along Spanish drove roads. These populations were located along the Conquense Drove Road (CDR; Fig. 1) in Spain, one of the few drove roads under continuous use since (at least) the Middle Ages until even after the post-industrialization crisis of Spanish transhumance (*Manzano Baena & Casas, 2010*). The CDR connects isolated grasslands inserted in the surrounding agricultural matrix, which act as excellent controls to check for effects of transhumance on landscape connectivity between plant populations. Under this scenario, a landscape genetics approach provides excellent tools to evaluate the effects of anthropogenic factors over the populations of *P. lagopus*. In particular, we asked: (i) Is the CDR activity influencing gene flow among populations? (ii) Do the populations in the CDR have similar genetic diversity and/or inbreeding values than those located in the agricultural matrix? (iii) Can other landscape factors (e.g., distance to human settlements, grasslands neighboring the drove road, time since last plowing) modify the genetic structure of the *P. lagopus* populations?

# MATERIALS AND METHODS

## Studied species

*Plantago lagopus* L. (Plantaginaceae) is a diploid polycarpic annual or biennial forb, abundant in grazed annual-dominated pastures of Central Spain (*Peco et al., 2005*). This species presents a persistent soil seed bank (*Peco et al., 2003*), gynodioecous breeding system and its seeds are dispersed by cattle endozoochory (*Malo & Suárez, 1995*). By sheep and under field conditions, epizoochoric dispersal is anecdotal, but endozoochory figures range from three to seven seeds per g dry weight of manure in late spring when dispersal is highest and flocks are moving northward, to 0.1 seeds in autumn when flocks are moving southward (P Manzano, 2004–2006, personal observations). This translates into ca. two million *P. lagopus* seeds moved northward per transhumance day, or 50,000 seeds moved southward, for a typical flock of 1,000 sheep. Experiments have also shown seeds are largely viable after sheep ingestion (*Peco, Lopez-Merino & Alvir, 2006*) and remain attached for a long time on sheep coats (*De Pablos & Peco, 2007*), even under real transhumance conditions (*Manzano & Malo, 2006*). This species is self-compatible, with both wind and insects as major pollination agents (*Sharma, Koul & Koul, 1993*). Annual-dominated pastures in Central Spain are known to present particularly high diaspore availability related to summer drought but translate into massive dispersal by
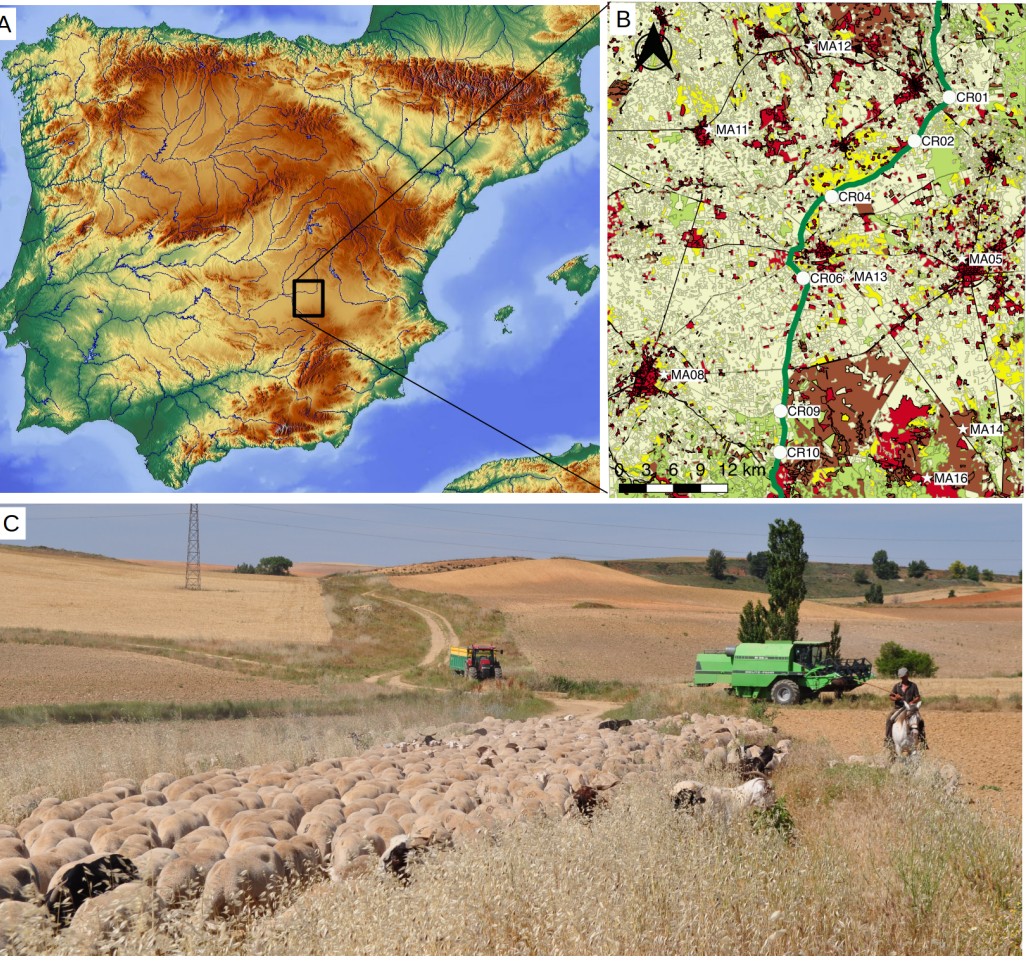

**Figure 1 Study area.** (A) Localization map of the sampled points within the Iberian Peninsula. Base map by maps-for-free.com. (B) Inset with location of sampling points on a simplified landcover map of the study area (elaborated with data compiled between 2009 and 2011 and provided by the National Geographic Institute at http://www.siose.es). Points were sampled in the drove road (circles, labeled "CR") and in the agricultural matrix (stars, labeled "MA"). The drove road ("Cañada Real Conquense"), depicted with broad green line, crosses a landscape formed predominantly by crops (pale yellow) and patches of woodlands (pale green), shrublands (brown) and grasslands (yellow). Urban and non-vegetated areas are shown in red. (C) Picture of the Conquense Drove Road between La Almarcha and Villar de la Encina in late spring, with transiting transhumant livestock herd. Note the longitudinal pasture with ripe grass, surrounded by agricultural fields. Image courtesy of José Antonio González.

livestock, particularly at the time when transhumant herds are on the move (*Manzano, 2015a*). Such large dispersal, along with the faster dynamics in annual plants (overlapping generations in perennial plants may disturb some genetic descriptors due to the mixture of genotypes), make them particularly suited to test genetic signals of dispersal in fragmented scenarios (*Ewers & Didham, 2006*).

## Landscape scenario and population sampling

The study area is located between Quintanar de la Orden, Tomelloso and Villarrobledo municipalities in Castilla-La Mancha, Spain (coordinates: 39°18′03.17″N and 2°49′54.50″W,

Fig. 1). The area is a plateau (830–900 m a.s.l.) of sandstones, loams and clay materials under continental Mediterranean climate, with a mean annual rainfall of about 500 mm and severe summer droughts. The vegetation is a mosaic of agricultural dry cereal and sunflower croplands, abandoned fields and dry grasslands used by local flocks. The area is crossed by the CDR, one of the major road droves (ca. 410 km long) that are still used for transhumant sheep and cattle herds that move every year from the cooler and wetter mountains of Teruel, Cuenca and Guadalajara provinces to the wintering dehesas in Sierra Morena at lower altitude (*Oteros-Rozas et al., 2012*). The herds cross the study area twice a year (northward in June and southward in November), currently comprising about 9,000 heads of sheep and 1,200 of cattle. The size of transhumant herds, dominated by merino sheep, has shrunk from about half a million in the 16th and 17th centuries to 100,000–200,000 in the next two centuries and around 20,000 during the second half of the 20th century (*Bacaicoa Salaverri, Elías Pastor & Grande Ibarra, 1993*). Productivity of the area is low except for peaks in May and October, because of cold winters and hot dry summers. Consequently, the numbers of transiting nomadic livestock have been historically much higher than those of resident livestock (*Manzano Baena & Casas, 2010*), which only grew in the last centuries (*Bacaicoa Salaverri, Elías Pastor & Grande Ibarra, 1993*). Such resident livestock herds are, in turn, kept at stables in urban settlements at night and graze grassland patches that are spread across the landscape (Fig. 1), overwhelmingly under private tenure. They graze mostly during wetter times in spring and autumn, the time of the year when local vegetation is growing (cf. Manzanares town in *Manzano Baena & Casas, 2010*) and diaspore availability is very low—in spring because plants have not yet produced seeds, and in autumn because rains incorporate the dead plant matter and any remaining diaspores into the soil. At other times of the year they are supplemented with hay, concentrate and water. Drove roads tend to avoid urban settlements and run rather tangentially (Fig. 1), so resident livestock do little use of them because of the rather radial pattern of land use, with urban settlements in the center. As a result, management of local resident livestock is likely to have some effect on pollen flow because of available pastures and to cause some short-distance dispersal across them, but it is not likely to have effects on long-distance seed dispersal along the drove roads.

In winter 2013 we selected 13 populations of *P. lagopus* (six in the CDR and seven in isolated pastures interspersed in the agricultural matrix). Populations located in the agricultural matrix were located at least five km from the CDR. Each population consisted of a group of plants not further than 100 m from each other, and all populations were at least five km apart from each other (Fig. 1; Table 1). For each population, we randomly collected between 8 and 30 flowering individuals of *P. lagopus*, whose leaves were stored under dry conditions until DNA extraction. Sampling was done without prior permission, in accordance with article 32d of the regional regulating law on drove roads regarding common complementary uses, including educational and formative activities in the environmental and cultural fields (*Junta de Comunidades de Castilla-La Mancha, 2003*). Authorization is required for such activities when they threaten the protection of sensitive ecosystem, forests with high wildfire risk, and protected flora and fauna species, and infringements are foreseen in its article 41—but our sampling in the ecosystem was

**Table 1  Location and genetic descriptors for studied populations.**

| Population | Coordinates | $N$ | $I$ | $A_P$ | $H_O$ | $H_E$ | $F_{IS}$_HW | RiqRare | PrivRare |
|---|---|---|---|---|---|---|---|---|---|
| LCR01 | 39.43 N, 2.62 W | 30 | 2.46 | 6 | 0.37 | 0.93 | 0.17–4 | 9.4 | 0.5 |
| LCR02 | 39.39 N, 2.67 W | 10 | 2.03 | 1 | 0.82 | 0.88 | 0.10–1 | 8.8 | 0.2 |
| LCR04 | 39.34 N, 2.77 W | 10 | 2.29 | 4 | 0.89 | 0.88 | 0.01–1 | 10.2 | 0.7 |
| LCR06 | 39.26 N, 2.81 W | 13 | 2.21 | 2 | 0.84 | 0.88 | 0.09–3 | 9.3 | 0.6 |
| LCR09 | 39.13 N, 2.84 W | 14 | 2.11 | 2 | 0.89 | 0.87 | 0.02–4 | 8.6 | 0.8 |
| LCR10 | 39.08 N, 2.84 W | 8 | 1.88 | 1 | 0.79 | 0.86 | 0.11–1 | 8.3 | 0.6 |
| *LCR average (CV)* | | | *2.16 (0.09)* | *2.7* | *0.76 (0.24)* | *0.88 (0.02)* | *0.08 (0.73)* | *9.1 (0.07)* | *0.6* |
| LMA05 | 39.27 N, 2.61 W | 10 | 2.21 | 3 | 0.85 | 0.89 | 0.06–2 | 9.8 | 0.5 |
| LMA08 | 39.16 N, 2.99 W | 30 | 2.49 | 12 | 0.78 | 0.84 | 0.14–4 | 9.6 | 0.8 |
| LMA11 | 39.41 N, 2.93 W | 14 | 2.29 | 1 | 0.86 | 0.89 | 0.11–2 | 9.4 | 0.3 |
| LMA12 | 39.49 N, 2.81 W | 10 | 2.04 | 5 | 0.89 | 0.89 | 0.1–3 | 8.8 | 0.8 |
| LMA13 | 39.26 N, 2.76 W | 13 | 2.47 | 1 | 0.88 | 0.92 | 0.09–3 | 10.7 | 0.6 |
| LMA14 | 39.11 N, 2.61 W | 10 | 1.42 | 1 | 0.75 | 0.86 | 0.17–2 | 5.1 | 0.4 |
| LMA16 | 39.06 N, 2.66 W | 14 | 1.99 | 3 | 0.84 | 0.91 | 0.11–2 | 7.9 | 0.4 |
| *LMA average (CV)* | | | *2.13 (0.17)* | *4.5* | *0.83 (0.06)* | *0.89 (0.03)* | *0.11 (0.35)* | *8.7 (0.21)* | *0.5* |

Note:

$N$ stands for number of individuals analyzed. Genetic diversity descriptors ($I$, Shannon information index; $A_P$, total number of private alleles; $H_O$, observed heterocigosity; $H_E$, expected heterozygosity), $F_{IS}$, inbreeding coefficient and number of alleles deviated from Hardy–Weinberg equilibrium are provided. RiqRare stands for rarefacted allelic richness, and PrivRare for average rarefacted number of private alleles. Average for drove road (LCR) and matrix (LMA) populations are also given.

negligible, and *P. lagopus* is neither an endangered nor a protected species. We deposited five individuals at Real Jardín Botanico de Madrid—CSIC herbarium (reference numbers MA-01-00892341–MA-01-00892345).

To test for the potential effect of other landscape factors on genetic descriptors, we measured the Euclidean distance from the sampling points to the nearest human settlement using Google Maps. At the agricultural matrix points, we also checked the time since last plowing by inspecting historic aerial photographs from the area (Instituto Geográfico Nacional, Spain)—the oldest of which being 70-years-old. The CDR, as in the case of all other major drove roads in Spain, has been protected from plowing for the last centuries (*Bacaicoa Salaverri, Elías Pastor & Grande Ibarra, 1993*).

### Microsatellite development and laboratory protocols

We developed specific microsatellite markers for *P. lagopus* in collaboration with the Unidad de Genómica, Parque Científico de Madrid (http://fpcm.es/en/servicios-cientificos/). Briefly, the library was developed from DNA of five individuals of *P. lagopus* extracted using QIAGEN plant Extraction Kit, using manufacturer's protocol. DNA was fragmented with ultrasounds using Bioruptor (Diagenode, Liège, Belgium) and size fragment was selected for an approximated size of 1.2 kb. DNA sequencing was performed using a 454-FLX-Plus Roche sequencer. We obtained approximately 185,000 sequences. We used Pal_Finder to select those sequences with more than five repetition motifs and a size that ranged from 100 to 300 bp. We used Primer3 (*Untergasser et al., 2012*) to configure potential primers that amplify those microsatellite regions. Details of the primers used are shown in Table S1.
## Genetic descriptors: diversity, inbreeding, differentiation and structure

We checked the six microsatellites used for genetic errors and misidentifications of alleles with Micro-checker (*Van Oosterhout et al., 2004*). We estimated four genetic diversity indices: observed and expected heterocigosity ($H_O$ and $H_E$) using INEST 2.1 (*Chybicki & Burczyk, 2009*), the Shannon information index (I), as a measure of genetic diversity that considers the number of alleles (NA) and its dominance in the population and the number of private alleles ($A_P$) using GenAlEx 6.5 (*Peakall & Smouse, 2012*). Because the number of sampled individuals is different in each population (ranging from 8 to 30, Table 1), we also calculated the genetic diversity using Hp-Rare (*Kalinowski, 2005*) to rarefact the allelic richness and the number of private alleles to the minimum number of individuals sampled in one population (i.e., eight individuals) for better comparisons between populations according to population size, hereafter $A_r$ and PrivRare, respectively. Linkage disequilibrium among loci was assessed with GenAlEx 6.5 (*Peakall & Smouse, 2012*).

We calculated the inbreeding coefficient, $F_{IS}$, using INEST 2.1. (*Chybicki & Burczyk, 2009*) that corrects the effect of null alleles, inbreeding and genotyping failures. DIC values between models with and without inbreeding were compared to evaluate the effect of inbreeding over $F_{IS}$. Departure from Hardy–Weinberg (HW) equilibrium with GeneAlEx 6.5 (*Peakall & Smouse, 2012*) to evaluate the inbreeding status of *P. lagopus* populations. We also assessed the presence of genetic bottlenecks generated by population dynamics (e.g., colonization events, population size reduction) using Bottleneck 1.2.02 (*Piry, Luikart & Cornuet, 1999*), through a Wilcoxon test with 2,000 permutations under the three plausible mutations models, as recommended for microsatellite data. We applied Bonferroni correction for multiple testing.

We estimated genetic differentiation through the $F_{ST}$ coefficient calculated with FreeNA (*Chapuis & Estoup, 2006*) to minimize potential biases caused by null alleles or allele dropout. We tested signification between each pair of populations with 999 permutations. We used HIERFSTAT 0.04 (*Goudet, 2005*) to evaluate the F-statistics grouping of the populations according to their location (either within the matrix or in the drove road) with 1,000 permutations. To assess the presence of isolation-by-distance (IBD) patterns, we performed a Mantel test with 999 permutations using genetic differentiation (i.e., $F_{ST}$ coefficients) and Euclidean geographical distance between populations with Vegan package in R (*Oksanen et al., 2007*). Mantel tests were also developed considering the location of the populations (Drove Road or Matrix) with the same permutations. Genetic differentiation between patches was compared with the calculation of migration rates or number of migrants. BayesAss (*Wilson & Rannala, 2003*) and Migrate (*Beerli, 2009*) were used to estimate recent and past migration rates, respectively, using the default parameters recommended by the manual. This might show differences among ancestral or recent gene flow, given the progressive decrease of livestock movement along the drove road during the last decades. Mantel tests were also developed considering the location of the populations (Drove Road or Matrix) with the same permutations.

We assessed genetic structure using the Bayesian clustering method STRUCTURE v. 2.3.4 (*Hubisz et al., 2009*) with prior information on populations. We carried out 10 independent runs for each $K$ value (i.e., number of groups or clusters), ranging from 1 to 15. Each run comprises a burning period of 100,000 permutations, followed by 1,000,000 Markov Chain Monte Carlo iterations. We assumed correlated allele frequencies and genetic admixture. To assess the most plausible number of clusters ($K$), we used the STRUCTURESelector module (*Li & Liu, 2018*) to obtain plausible values of $K$. We considered the Evanno method (*Evanno, Regnaut & Goudet, 2005*) with the ad hoc statistic $\Delta K$ and identified the maximum $\Delta K$ when plotted against $K$. MedMeaK (median of means), MedMedK (median of medians), MaxMeaK (maximum of means) and MaxMedK (maximum of medians) following *Puechmaille (2016)* were also estimated (threshold value $= 0.6$). These estimators are useful to discard spurious clusters and detect substructure in case of uneven sampling (*Puechmaille, 2016*). We also used Clump v.1.1.2 (*Jakobsson & Rosenberg, 2007*) to estimate the membership coefficient of each individual in each cluster and Distruct v 1.1 (*Rosenberg, 2004*).

## Relationships of genetic descriptors with environmental factors

To assess whether movement of transhumant herds along the drove road leaves a footprint, we selected the genetic descriptors that have most commonly been used previously in similar studies, i.e., $F_{IS}$ and $A_r$ (*Rico, Boehmer & Wagner, 2014a*; *Rico et al., 2014b*; *Rico & Wagner, 2016*; *DiLeo et al., 2017*; *Plue, Aavik & Cousins, 2019*). We built multiple linear models fitted with generalized least squares (*gls* function, *nlme* R-package) to examine the relationship between genetic diversity metrics with the population location (agricultural matrix or drove roads), the distance to nearby human settlements (in km) and the interaction location $\times$ distance.

Due to the effect of pollinators activity in the crossing system (i.e., selfing, partial selfing or outcrossing, *Turner, Stephens & Anderson, 1982*; *Dutech et al., 2005*; *Van Etten et al., 2015*), for the analysis of $F_{IS}$ the amount of grasslands in 500-m radius circles centered on sampling points (in percentage, log-transformed to achieve normality) was also included. The landscape descriptors (distance and percentage of grasslands) were standardized and centered to interpret model coefficients as standardized effects. Spatial autocorrelation was taken into account by building the GLS models with three alternative spatial covariance structures most commonly used to describe spatial correlation (Gaussian, Exponential and Spherical). The presence of nearby grasslands was included for the analysis of $F_{IS}$, for this genetic descriptor is considered to be particularly influenced by pollinator activity due to its effect over selfing and/or kinship crosses (*Rico et al., 2014b*). Long-distance seed dispersal will rather happen along drove roads and through transhumant livestock that bypass such nearby grasslands, because the resident livestock that use them tend to have short-distance movements around human settlements and do not use the drove roads that run rather tangentially to the settlements.

The differences in genetic diversity among populations may show a nested pattern, suggesting that former connections between them have been lost (*Aggemyr & Cousins, 2012*; *Plue, Aavik & Cousins, 2019*). Nestedness appears when the set of alleles in

communities with lower genetic diversity are a subset of those found in communities with higher genetic diversity, which would imply a deterministic loss of alleles. Thus, we explored whether such nested patterns occurred among our study populations and if they could be linked to the presence of the drove road, the distance to settlements and the amount of neighboring grasslands. If nestedness among populations is caused by loss of connectivity, we expect sites within the agricultural matrix and far from settlements to have lower allelic richness, lacking some particular alleles present in the populations of the drove road (in other words, sites within the agricultural matrix would have a higher degree of nestedness). Neighboring grasslands could act as a local source of propagules (seed and pollen) deterring the loss of alleles, and thus the larger the availability of neighboring grasslands the less likely we would find nestedness patterns. To describe nestedness patterns we used the NODF index (*Almeida-Neto et al., 2008*). We tested if nestedness with a suitable null model based on 499 randomized matrices (null model "r2" with package vegan in R which constrains row totals –allelic richness—to equal the observed distribution and samples columns –alleles—with probability proportional to squared incidence frequencies). Then, we modeled the population rank order, from the highest rank of the population with most alleles to the lowest rank of the population with least alleles, with linear models built with GLS as above. The populations differed in the number of individuals analyzed. Then, to balance sampling effort among them we repeated 50 times the above procedure, by randomly selecting the same number of individuals per population. Final NODF, *p*-values and rank estimates were the mean of the above procedure for these 50 balanced replicates.

For the three genetic diversity metrics ($F_{IS}$ and $A_r$) and for nestedness, we did not try to simplify the models but to identify the relevant predictors with *p*-values based on nested log-likelihood ratio tests. The genetic descriptors we used as response variables in the models were the inbreeding coefficient ($F_{IS}$), the genetic diversity statistic ($A_r$) and the nestedness estimates. We confirmed nearby grasslands to be only a relevant factor for $F_{IS}$ by comparing models of the three metrics with and without the grassland factor via AICc. Additionally, we tested the correlation of the genetic descriptors with the plowing history (years since last plowing) with base R and *Anova* function, *car* package and just for the populations in the agricultural matrix—the drove road having been protected from plowing for the last centuries.

## RESULTS

### Genetic diversity and inbreeding

The values for genetic diversity ($I$, $A_P$, $H_O$, $H_E$, $A_r$ and PrivRare) were similar between populations in the agricultural matrix and populations in the drove road, except for $A_P$, which was greater in the populations of the agricultural matrix, due to the large NA found in LMA08 (Table 1). The average (CV) expected heterozygosity $H_E$ was 0.88 (0.02) in drove roads populations and 0.89 (0.03) in matrix populations. However, matrix populations showed greater variability in $H_E$, with values ranging from 0.92 in LMA13 to 0.84 in LMA08. The same pattern (similar average but different variability) was found in the other genetic diversity indexes (Table 1).

**Table 2 Bottleneck output for heterozygosity excess (population bottlenecks) in 13 studied populations of *Plantago lagopus*.**

|        | IAM  | TPM  | SMM  |
|--------|------|------|------|
| LCR01  | 0.43 | 0.84 | 0.11 |
| LCR02  | 0.57 | 0.57 | 0.84 |
| LCR04  | 0.15 | 0.43 | 1    |
| LCR06  | 1    | 0.43 | 0.15 |
| LCR09  | 0.15 | 0.15 | **0.01** |
| LCR10  | 1    | 0.84 | 0.84 |
| LMA05  | 0.68 | 1    | 0.11 |
| LMA08  | 0.56 | 0.56 | **0.03** |
| LMA11  | 0.43 | 0.56 | **0.03** |
| LMA12  | 0.43 | 0.43 | 1    |
| LMA13  | 0.56 | 1    | 0.84 |
| LMA14  | **0.01** | 0.11 | 0.56 |
| LMA16  | 0.68 | 0.84 | 0.15 |

**Note:**
The output is provided under three evolutive models: infinite allele model (IAM), two-phase model (TPM) and stepwise mutation model (SMM). Significant values after Wilcoxon sign-rank test with Bonferroni correction are highlighted in bold. Population code are belong into drove road (LCR code) or matrix (LMA code) populations.

The inbreeding coefficient ($F_{IS}$) varied greatly among populations and between matrix and drove road locations. Drove road populations showed a mean value of 0.08 (ranging from 0.01 in LCR04 to 0.17 in LCR01), while matrix populations showed a mean value of 0.11 ranging from 0.06 in LMA05 to 0.17 in LMA14 (Table 1). Inbreeding had an important effect over *Fis* in LCR01, LMA11, LMA14 and LMA16 populations according to INEST model estimation. All populations had, at least, one locus that deviated from HW equilibrium (four loci in LCR01). Bottleneck signals were found in some populations under SMM model (LCR09, LMA08 and LMA11), but the results were not significant in all genetic models and when applying multi-comparisons corrections (Table 2). Linkage disequilibrium among loci showed only four significant comparisons, which were non consistent between loci pairs.

## Genetic differentiation and structure

Pairwise genetic differentiation values ($F_{ST}$) between populations were low (Table S2). On average, LMA14 showed the greatest differentiation, with an average $F_{ST}$ value of 0.08. Hierarchical analysis of the genetic differentiation showed low differentiation between matrix and drove road locations ($F_{CT} = 0.032$) and a moderate one between populations within each category ($F_{SC} = 0.11$). Both values are significant. Mantel test ($r = 0.13$; $p = 0.22$) did not detect the presence of an IBD pattern, but results were significant when only analyzing the six populations included in the drove road ($r = 1$; $p < 0.01$). The genetic structure analyzed with a Bayesian clustering approach (i.e., STRUCTURE, Fig. 2) showed different values depending of the approach consider to estimated K. The estimation of ∆K proposed values of $K = 12$ and $K = 4$ as more plausible approaches. MedMedK, MedMeanK and MaxMaxK also pinpoint that $K = 4$ is the most appropriate

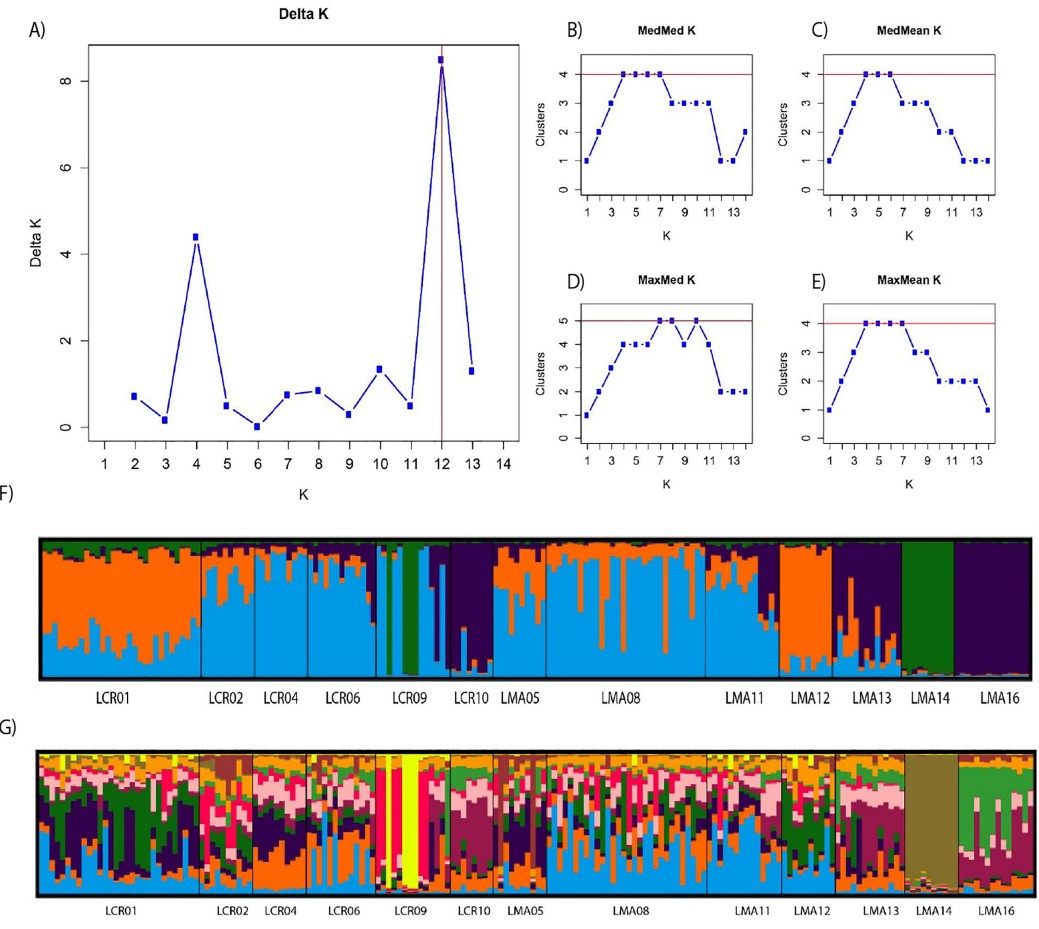

**Figure 2 Bayesian genetic structure results obtained with STRUCTURE.** Number of clusters proposed by Evanno method (A). Number of clusters proposed MedMedK (B), MedMeanK (C), MaxMedK (D) and MaxMeanK (E). Genetic structure proposed for $K = 4$ (F). Genetic structure proposed for $K = 12$ (G).

distribution of the genetic structure and MaxMedK suggested that $K = 5$ is the most accurate result. The $K = 4$ distribution showed an independent cluster (green) for LMA14 population, another cluster (purple) for LMA16, LMA13 and LCR10 populations and a third cluster (orange). The other populations are assigned to the last cluster, with admixture of the others. The $K = 12$ distribution showed an important population admixture and one cluster assigned to LMA14 population (Fig. 2). The results for gene flow (BayesAss and Migrate, Table 3; Fig. 3) showed differences for the migration rates estimated, with higher values for past gene flow (i.e., Migrate) than contemporary (i.e., BayesAss). The evaluation of the recent gene flow (i.e., BayesAss), focus the main differences in the population LMA08 that received a significant number of migrants and in lesser extent, LMA16 and LCR01. The remaining populations showed similar migrations rates between populations. The ancient gene flow analyzed (i.e., Migrate) showed similar mean values for drove roads and matrix populations but important differences between populations. Although LMA08 maintains high values of gene flow, LCR02 received the highest average migration rate meanwhile LCR10 showed the smallest average value.

**Table 3 Summary of the average rate (± standard deviation) of gene flow received from each population.**

| Population | BayesAss | Migrate |
|---|---|---|
| LCR01 | 0.021 ± 0.016 | 16.77 ± 5.69 |
| LCR02 | 0.012 ± 0.003 | 20.23 ± 6.29 |
| LCR04 | 0.012 ± 0.003 | 16.96 ± 8.77 |
| LCR06 | 0.013 ± 0.003 | 16.58 ± 8.13 |
| LCR09 | 0.013 ± 0.003 | 12.58 ± 4.99 |
| LCR10 | 0.013 ± 0.002 | 6.95 ± 4.13 |
| LCR average | 0.014 ± 0.003 | 15.01 ± 4.63 |
| LMA05 | 0.013 ± 0.002 | 10.64 ± 5.47 |
| LMA08 | 0.11 ± 0.067 | 20.03 ± 9.18 |
| LMA11 | 0.013 ± 0.003 | 17.04 ± 6.76 |
| LMA12 | 0.013 ± 0.003 | 15.58 ± 7.07 |
| LMA13 | 0.013 ± 0.003 | 13.08 ± 6.16 |
| LMA14 | 0.013 ± 0.003 | 14.91 ± 7.36 |
| LMA16 | 0.04 ± 0.03 | 17.36 ± 9.88 |
| LMA average | 0.031 ± 0.036 | 15.52 ± 3.06 |

**Note:**

The gene flow is estimated with BayesAss (m) and Migrate ($N_e$m) software. Average ± SD for location (drove road and matrix) have been also calculated.

## Landscape factors governing genetic descriptors in *P. lagopus* populations

The distance to the nearest settlement had an impact on both genetic descriptors analyzed (Table 4). This relationship is independent of the frequency of plowing, because for the agricultural matrix plots the correlations of years since last plowing and the residuals of the regressions explaining the genetic descriptors were non-significant. Also, the values of genetic descriptors did not correlate significantly with the number of years since last plowing (Pearson correlations were 0.07, $p = 0.89$ for $F_{IS}$; and $-0.13$, $p = 0.78$ for $A_r$).

In addition to the distance to the nearest settlement, the interaction with population position showed a significant effect on inbreeding ($F_{IS}$; Table 4). Thus, $F_{IS}$ increased with distance from human settlements in the agricultural matrix, but followed the opposite trend in drove road populations (Fig. 4A). The presence of nearby grasslands also proved to be a significant factor, showing higher rates of inbreeding at higher grassland cover (Table 4).

For the descriptor of genetic diversity ($A_r$), the distance to settlements had a consistent negative and significant effect, while the effect for population position and its interaction with distance was only marginally significant (Table 4). $A_r$ values between matrix and drove road positions showed a pattern of convergence at short distances and divergence at high distances from human settlements, with the agricultural matrix then showing less diversity (Fig. 4B).

Significant nestedness was found among the *P. lagopus* populations (mean observed NODF = 34.7, mean *p*-value = 0.02, all averaged from 50 random selections of individuals

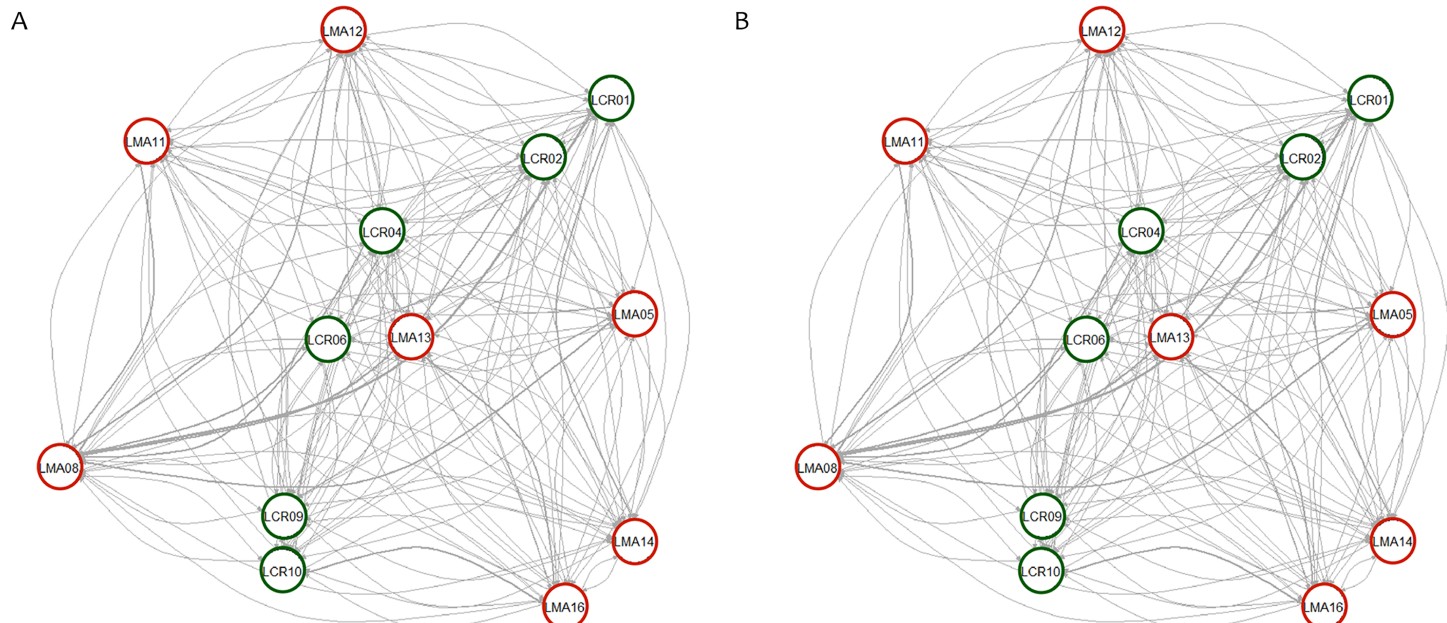

**Figure 3 Contemporary (BayesAss) (A) and historical (Migrate) (B) rates of gene flow among the *Plantago lagopus* populations.** Matrix (LMA) populations are represented with red circles, while Cañada (LCR) populations are surrounded by green circles. The width of the edges is proportional to $m$ and $N_e m$ values, respectively.

to balance sample size among populations). None of the geographic parameters, however, significantly affected the rank (Table 4).

## DISCUSSION

Drove roads have a significant capacity to shape some genetic features of *P. lagopus* populations in the study area. This assessment of the population genetics of *P. lagopus* in the study area indicates relevant differences both in terms of coefficient of inbreeding and of genetic diversity between populations and between matrix and drove roads positions. Furthermore, proximity to human settlements appeared to drive the hotspots for genetic diversity in this landscape. Drove roads were able to invert the inbreeding that otherwise appears in distant populations from the agricultural matrix. The loss in genetic diversity with increasing distance to human settlements seems weaker at drove roads, although such effect is only marginally significant with the sample size and the molecular markers used in this study. Such a trend would, however, be in line with the one that was observed by other studies involving livestock being moved on hoof (*Rico, Boehmer & Wagner, 2014a*; *Rico & Wagner, 2016*; *DiLeo et al., 2017*), as well as by more artificial transport (boat; *Plue, Aavik & Cousins, 2019*). The higher estimated migration rates for past events compared to contemporary ones would confirm the declining influence of transhumance in the landscape as transhumant livestock numbers continue to decline. An influence of historical processes of colonization can be discarded, due to the absent effect that plowing history had both on inbreeding and diversity on the populations from the agricultural matrix—the time span considered for plowing history, i.e., 70 years, should also allow for controlling delay effects related to the establishment of a permanent

**Table 4 Results of linear modeling of the inbreeding depression ($F_{IS}$), rarefacted allelic richness ($A_r$) and nestedness on landscape factors.**

| Genetic descriptors | | Coefficient estimate | $p$ |
|---|---|---|---|
| $F_{IS}$ | $\chi_5^2 = 20.82$, $R^2 = 26\%$, Gaussian spatial covariance structure | | **0.001** |
| | Intercept | 0.139 (0.028) | |
| | Distance to settlement | 0.015 (0.032) | **<0.0001** |
| | Position (drove road) | −0.015 (0.025) | 0.280 |
| | Grassland cover | 0.037 (0.009) | **<0.0001** |
| | Distance × Position | −0.106 (0.037) | **0.005** |
| $A_r$ | $\chi_4^2 = 10.51$, $R^2 = 72\%$, Gaussian spatial covariance structure | | **0.033** |
| | Intercept | 8.056 (0.459) | |
| | Distance to settlement | −1.724 (0.548) | **0.006** |
| | Position (drove road) | 1.254 (0.666) | 0.076 |
| | Distance × Position | 1.282 (0.705) | 0.070 |
| Nestedness | $\chi_4^2 = 11.68$, $R^2 = 29\%$ | | **0.019** |
| | Intercept | 5.620 (1.354) | |
| | Distance to settlement | −3.711 (1.558) | 0.416 |
| | Position | 2.470 (1.894) | 0.404 |
| | Distance × Position | 1.094 (2.420) | 0.160 |

Note:
The linear predictor is built with the population position (drove road/agricultural matrix), the distance to the nearest settlement, the availability of neighboring grasslands (only for $F_{IS}$; log-transformed) and the interaction between position and distance. Landscape descriptors were standardized and centered. Models were fitted using generalized least squares (GLS) with three alternative spatial correlation structures (Gaussian, exponential, spherical) for the residuals, to account for potential spatial autocorrelation. We show here the best models according to AICc. The log-likelihood ratio test ($\chi^2$) is given after comparison with a null model with the same spatial structure and corresponding $p$-value, as well as an approximation to $R^2$ as the squared Pearson correlation between predicted and observed values, and coefficient estimates (with standard errors) and $p$-values from a type-II sum of squares for the explanatory variables (note that coefficient for "Position" factor refers to "drove road" level, while the accompanying $p$-value is for the factor as a whole). Significant $p$-values ($p < 0.05$) are highlighted in bold.

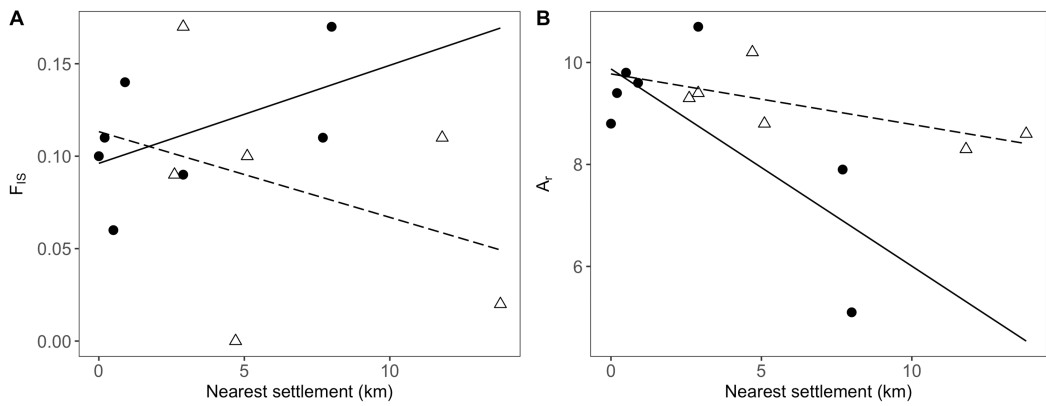

**Figure 4 Relationship between genetic and geographic parameters.** Linear regressions are shown between distance to closest settlement for populations in the agricultural matrix (solid circles and solid line) and those in the drove road (open triangles and dashed line), and genetic descriptors: (A) $F_{IS}$ for inbreeding and (B) $A_r$ (rarefacted allelic richness) for diversity.

seed bank as discussed by *Honnay et al. (2008)* and *Plue et al. (2017)*, of which *P. lagopus* is capable.

In a self-compatible plant species pollinated both by wind and by insects, such as *P. lagopus*, population inbreeding is mainly determined by the ratio between selfing and outcrossing, where outcrossing is enhanced by pollinator availability. Thus, the smaller selfing coefficient and the reduced influence over $F_{IS}$ observed along the drove road indicates a relevant role played by them in providing pollination services. This relationship is even clearer after controlling for landscape effects, namely the presence of grassland patches in the vicinity of the studied plant populations that could harbor further pollinators. In fact, we found the same potential negative relationship between the presence of such patches and the pollinators, mirroring the results of *Rico et al. (2014b)* and, similarly to the potential explanation there, it could rely on the flowering plants on such patches being continuously grazed during the whole spring by resident sheep flocks. Meanwhile, the transhumant herds graze on the drove road only at the end of the spring, following a "green wave" when the plants have already grained, relatively untouched during the flowering phase. While a greater coefficient of inbreeding is a genetic indicator that can be related to pollinator limitation (*Turner, Stephens & Anderson, 1982*; *Dutech et al., 2005*; *Van Etten et al., 2015*), the structural role of drove roads as grassland corridors crossing agricultural landscapes has been proved to be determinant in supporting pollinator services (*Hevia et al., 2016*), which our study confirms. This result goes along other studies that have observed provision of heterogeneity at the landscape level because of the drove road's structural role, translated in higher biodiversity levels (*Azcárate et al., 2013a*; *Hevia et al., 2013*). Plowing does not show any effect or trend on the genetic variables studied, suggesting that structural continuity in space is more important than stability in time to guarantee genetic admixture.

Distance to human settlements showed a consistent effect on genetic diversity and inbreeding. The resulting functional landscape can be interpreted as settled, high human density areas being hotspots of dispersal that modulate the genetic patterns of diversity, probably mediated by commercial livestock exchanges but also by other dispersal types, mainly human-mediated (*Auffret, 2011*; *Auffret & Cousins, 2013*; *Wichmann et al., 2009*). The *P. lagopus* populations on the drove road don't show to be affected by these types of dispersal, likely because their human use is much diluted and human activities are comparatively simplified and less dense, consisting just in accompanying or herding livestock. They also show to be pollinator reserves with a higher relevance than peri-urban grassland patches. Such important role played by human settlements for seed dispersal is not surprising, given that human-mediated dispersal adds to livestock-mediated dispersal mechanisms (*Auffret, 2011*; *Auffret & Cousins, 2013*) which are problematic for the spread of invasive species (*Abbas, Mancilla-Leytón & Castillo, 2018*). However, it offers insights on how an unnatural over-dispersal that facilitates invasiveness of some plant species (*Wilson et al., 2009*) may be visible already at small geographical scales such as the one regarded in this study. Genetic analyses that further explore the role of livestock or other dispersal vectors may offer valuable insights on the ecology of potential or confirmed invasive species, especially in relationship with their human-mediated dispersal.

The trend of a higher genetic diversity on drove roads compared to the agricultural matrix (Fig. 4B), even if only marginally significant and limited by the reduced sample size and the limited number of molecular markers, would be consistent with the determinant role of livestock as seed disperser at the genetic level that has been observed in other studies (*Rico, Boehmer & Wagner, 2014a*; *Rico & Wagner, 2016*; *DiLeo et al., 2017*; *Plue, Aavik & Cousins, 2019*). It may be pointing to a scenario where two types of dispersal processes coexist. On the one hand, populations in the agricultural matrix would receive seeds through the local stock through short-distance dispersal processes. Meanwhile, long-distance dispersal processes favoring greater admixture dominate in the drove roads actively being used by livestock, therefore functioning as a highway for seeds. This would highlight the important role of active livestock movements for preserving the functionality of drove roads and, more generally, herbivore migration corridors. Such effects seems to be too weak to be perceived as significant, given our low sample size and reduced molecular markers used, on top of the moderate levels of gene flow that homogenize the genetic diversity between the populations. The observable trend and the estimated effects on past and current migration rates, however, may be pointing to a very promising area of research for future studies, especially considering that the long history of drove roads in Spain, with fixed ancient routes coinciding with older herbivore routes, may be showing a sustained contribution to long-distance seed dispersal (*Manzano Baena & Casas, 2010*; *Manzano & Malo, 2006*). Further insights on the persistence of effects may be gained by comparing the CDR with other Spanish drove roads abandoned in the mid-20th century (*Manzano Baena & Casas, 2010*).

## Implications for ecosystem function and conservation

Functional herbivore corridors would sustain diversity by reducing the risk of local extinctions due to increased inbreeding and loss of genetic variation in small populations (*Caughley, 1994*). The effect of drove roads on inbreeding observed in our study has relevant implications for conservation, as inbreeding depression is a major concern for the conservation of biodiversity in human-altered fragmented landscapes (*Hedrick & Garcia-Dorado, 2016*). We think that connectivity factors, related to herbivore type and capability of sustaining dispersal corridors for plants, as well as optimal matching of grazing times with the flowering and seeding phenology of plants, would therefore add to the complexity of factors that influence the relationship between grazing and plant diversity. Such factors include productivity and disturbance (*Herrero-Jáuregui & Oesterheld, 2018*; *Olff & Ritchie, 1998*), historical effects on communities (*Cingolani, Noy-Meir & Díaz, 2005*; *Osem, Perevolotsky & Kigel, 2002*) or herbivore species and sizes (*Bakker et al., 2006*; *Liu et al., 2015*). Functional herbivore corridors would sustain diversity by reducing the risk of local extinctions due to loss of genetic variation in small populations (*Caughley, 1994*).

The marginally significant, yet observable trend of drove roads in reducing genetic diversity loss of *P. lagopus*, in spite of the rapid transit of transhumance flocks—ca. 15 km/day or 1.5 km/h when moving (*Hevia et al., 2013*) is very noteworthy. Such a use, which is intense but nevertheless very restricted in time, would achieve extraordinary

dispersal effects due to a "green wave effect" (*Merkle et al., 2016*), as transhumant flocks use drove road pastures when they are at their maximum productivity (*Manzano Baena & Casas, 2010*) and therefore at their maximum seed production stage—which would add to the reduced impact on pollinators we observed that is also related to the "green wave effect." This is a radical contrast with other works studying rotational grazing systems (*Rico, Boehmer & Wagner, 2014a*; *Rico et al., 2014b*; *Rico & Wagner, 2016*; *DiLeo et al., 2017*; *Plue, Aavik & Cousins, 2019*) where the intensity of use per unit of area is much higher and the chances for consolidating dispersal between patches are also higher, and the effects on pollinator community are less clear. Indeed, a confirmation of such trend in future studies that are able to study more populations than we did would highlight the potential role of herbivore corridors to promote long-distance dispersal. The higher estimated migration rates in the past than in the present are also an alert on the negative effect that declining transhumant practices may have on ecosystems.

Mobile pastoralism is globally threatened by non-conductive policies (*Manzano, 2017*), weakening its role as a provider of important ecosystem functions such as habitat provision, nutrient cycling or bush encroachment/fire control (*Leroy et al., 2018*). The potential magnitude of the threat is great, for such land use covers more than half of the global land (*Manzano, 2015b*). A derived disruption of dispersal processes and the structural role for pollinators played by corridors themselves may further contribute to a generalized grassland biodiversity crisis motivated by land use intensification (*Gossner et al., 2016*) and climate change—and on the latter, a potential decoupling of herbivore "green wave" movements from the phenology and seed availability brings also concerns (*Berg et al., 2010*; *Thackeray et al., 2016*). It is urgent to quantify such dispersal beyond our study in order to orient adequate science-evidenced policy recommendations (*Manzano Baena, 2012*). Furthermore, corridors per se are important beyond livestock. We obtained our data in a drove road that has experienced a drastic decline in use in the recent centuries, from 600,000 head in the 16th century to less than 10,000 nowadays (*Bacaicoa Salaverri, Elías Pastor & Grande Ibarra, 1993*; *Hevia et al., 2013*). The effect on the genetics of plants must have been much more significant at that time, similar to what they must be in the systems presently steered by hundreds of millions of mobile pastoralists (*Manzano & Agarwal, 2015*), whose capacity to cover wide territories is shown by the fractal structure of their drove roads in different countries (*Manzano-Baena & Salguero-Herrera, 2018*). This effect should also be greater in the world's greatest wild herbivore migrations (*Berger, 2004*).

## CONCLUSIONS

Our results are relevant for shaping conservation policies that take into account the role of herbivores, be them wild or domestic. We confirm the relevance that herbivore corridors and mobile pastoralism have in reducing inbreeding levels in plant populations, as well as their potential, within the possibilities of our limited data, in facilitating long-distance plant migration that has been envisioned in seed dispersal studies (*Manzano, Malo & Peco, 2005*; *Manzano & Malo, 2006*). Such corridors may have to be kept in use in order to preserve full ecological functionality (*Starrs, 2018*) that goes

beyond their role as landscape structures. Their potential adds to the already identified needs to reduce fragmentation in the dryland areas occupied by pastoralists (*Said et al., 2016*), and international conventions that deal with such issues (*Durant et al., 2015*). Some hotly debated national policies related to drove roads (*AA.VV., 2012*; *Herzog et al., 2005*) should as well take note on our results. We also highlight the interest of expanding such studies to other animal-dispersed species in order to quantify the importance of corridors for the maintenance of population genetic exchange levels in a world confronted with global change, and where deleterious effects on plant genetic diversity are already visible (*Alsos et al., 2012*).

## ACKNOWLEDGEMENTS

P. M. would like to thank F. Xavier Picó Mercader for an enlightening dinner conversation at the 2nd Iberian Ecological Congress in July 2006 that was key to boost many of the ideas in this paper, as well as to Juan Orellana Saavedra and Raquel Casas Nogales for previous design attempts of this study, and Daniel McGahey for revising the English of the abstract. Juan Carlos Illera offered invaluable guidance during the initial stages of genetic analyses. Ricardo Ramos, Jesús García and Sara Álvarez from the genomic units at Cantoblanco and Moncloa of the Madrid Science Park (https://fpcm.es) provided the PCR and sequencing services with a great amount of flexibility.

### Funding

Financial support was provided by the Spanish MINECO (Projects CGL2011-24871, CGL2014-53789-R and CGL2016-77377-R) and the Madrid Regional Government (Project REMEDINAL-3). Financial support has also been provided by Instituto de Estudios "Don Juan Manuel" (Diputación de Albacete). The funders had no role in study design, data collection and analysis, decision to publish, or preparation of the manuscript.

### Grant Disclosures

The following grant information was disclosed by the authors:
Spanish MINECO: CGL2011-24871, CGL2014-53789-R and CGL2016-77377-R.
Madrid Regional Government: REMEDINAL-3.
Instituto de Estudios "Don Juan Manuel" (Diputación de Albacete).

### Competing Interests

The authors declare that they have no competing interests.

### Author Contributions

- Alfredo García-Fernández performed the experiments, analyzed the data, contributed reagents/materials/analysis tools, prepared figures and/or tables, authored or reviewed drafts of the paper, approved the final draft.
- Pablo Manzano conceived and designed the experiments, analyzed the data, prepared figures and/or tables, authored or reviewed drafts of the paper, approved the final draft.

- Javier Seoane conceived and designed the experiments, performed the experiments, analyzed the data, prepared figures and/or tables, authored or reviewed drafts of the paper, approved the final draft.
- Francisco M. Azcárate conceived and designed the experiments, performed the experiments, authored or reviewed drafts of the paper, approved the final draft.
- Jose M. Iriondo contributed reagents/materials/analysis tools, authored or reviewed drafts of the paper, approved the final draft.
- Begoña Peco conceived and designed the experiments, authored or reviewed drafts of the paper, approved the final draft.

### DNA Deposition

The following information was supplied regarding the deposition of DNA sequences:

The sequences developed here for six microsatellites for *Plantago lagopus* are accessible via GenBank accession numbers MF490249–MF490254.

### Data Availability

The raw genotyping data of Plantago are available in the Supplemental Files.

### Supplemental Information

Supplemental information for this article can be found online at http://dx.doi.org/10.7717/peerj.7311#supplemental-information.

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
