# Peer review of "Herbivore corridors sustain genetic footprint in plant populations: a case for Spanish drove roads"

_PeerJ, doi:10.7717/peerj.7311_

## Round 0.1 · original submission · Major Revisions

I have now received two very detailed reviews of your work. As you can see both reviewers are supportive of your paper but request a major revision.

Although Reviewer 2 is concerned with the use of only six microsatellite markers, they seem to show enough polymorphism to ensure sufficient statistical power. However, I would like to see all loci found in linkage disequilibrium or showing strong deviations from the HWE removed from the analyses. This was not done, and it could cause important biases in subsequent analyses. Reviewer 2 also requests to improve STRUCTURE analyses and account for the incidence of null alleles.

Both reviewers suggest you incorporate additional landscape metrics such as landscape conductance and landscape heterogeneity, and request you perform formal landscape genetic analyses (like MLPE models) to explicitly relate genetic differentiation to landscape attributes. I also encourage you to run MLPE models, which are becoming a standard method in landscape genetic studies. Additionally, regression models of genetic diversity metrics should explicitly account for spatial autocorrelation, as stated by Reviewer 1.

Finally, both reviewers stress out that you cannot distinguish between pollen-mediated gene flow and seed dispersal, and that pollen-mediated gene flow has not been properly addressed throughout the paper. They also state that all analyses need proper justification and should address the study’s initial objectives, and that the discussion should be focused to the results supported by your data. Speculation is valid, but should be identified as such. You are finally encouraged to use a land cover map in Figure 1 to explicitly show the location of the drove roads and land use types in the surrounding matrix.

·

Basic reporting

I think the start of the introduction, L. 47-59, should be removed. This is really besides the point of your manuscript and is not even tangentially relevant to what your study investigates.

As a general comment to improving the introduction and putting it more in line with what is investigated (importance of drove road vs. landscape characteristics), I may suggest using the framework of plant functional connectivity (in the drove road), as defined by Auffret et al. (2017) in Journal of Ecology, held against the role of structural connectivity (landscape characteristics). This will make the introduction more in touch with relevant theoretical advances in the field and open a good body of literature which will strenghten the introduction. In line with that, why did the authors only consider distance to human settlements as a quantification of landscape factors? There are other landscape characteristics which could be tested such as landscape heterogeneity around each population? On a side note, time since last ploughing is no landscape factor, and since the factor has no relevance, it could be omitted and replaced by factors such as landscape heterogeneity.

The authors should read the collective works of Yessica Rico (beyond the single one cited in the ms) and Michelle DiLeo, including my own work soon to be published: Plue, Aavik and Cousins, Diversity and Distributions (in press, deep link doi.org/10.1111/ddi.12842), for a broader empirical framing of the authors work. Using these papers, it may also be relevant in both the introduction and discussion as to how our collective works may lead to valuable generalizations of plant functional connectivity provided by domestic animal vectors.

In addition, the authors focus consistently on seed flow, but essentially ignore the role of pollen flow in the introduction, as to how that factor is key in outlining spatial population genetic structure, often considered more important than seed-based gene flow. This should be properly incorporated throughout the manuscript. It will help highlight why we should care about looking into seed-based gene-flow as well.

Experimental design

The RQ on genetic differentiation comes out of the blue. Needs a proper introduction beyond the statement on L. 132.

Methods comments

The sampling effort at 13 populations is at the lower end of what could be acceptable for a landscape genetic study, and this is made somewhat worse by the fact that half of the populations are below the 25-30 individuals which is the recommended sampling density for accurately assessing genetic diversity using microsatellites. Also, the authors equally only developed 6 markers, though 10 is commonly strived towards (i.e. the number used by journals to allow publication on the development of SSRs), to ensure a high enough resolution to detect the genetic patterns they are looking for. The authors acknowledge some shortcomings (at least the low number of populations, see L. 368), but I think they should do so more openly on all shortcomings and discussing the various methodological limitations more in-depth.

Why no calculations of raw A or effective allelic richness (Ae) values?

Many data analysis seem to have been executed without a clear rationale in relation to the research questions. Why the bottleneck analysis? Why the IDB analysis? Why the in-depth analysis of population genetic structure via STRUCTURE? The latter makes sense, but it is not clear from the introduction, whereas the former appear executed because they represent standard analysis in population genetic studies, but not because of a clear reason why they are relevant to this study. If they are, elaborate here and elsewhere in the manuscript! If not, omit! If an in-depth analysis of spatial population structure based on Fst is desired, it needs to be properly introduced (so more than L. 132) and related to the paper’s main themes and the analyses itself should be updated, beyond the much-criticized Mantel tests. For suggestions on state-of-the-art approach in doing so, see Plue, Aavik and Cousins in Diversity and Distributions (in press, deep link doi.org/10.1111/ddi.12842).

The statistical analysis needs (much) more detail. Also, one comprehensive GLS-model which incorporates all environmental variables would be more effective and can work on the raw rather than the rarified data by correcting for sample size, e.g., in R-code,

m1 <- gls(Fis ~ (location + distance + ploughing + sample size)^2, correlation = corGaus(form = ~X+Y, nugget = T), method ="ML")

The AMOVA could be replace by a similar more updated technique:

m1 <- gls(fst.dist ~ (location + distance + ploughing + sample size)^2, method = "ML", correlation = corMLPE(value = 0.1, form =…)).

Note that both analyses enable a correction for spatial autocorrelation via the “correlation” term in the model.

Validity of the findings

Results

I would like the authors to start with some summary stats on the genetic population descriptors across loci, populations, etc. To help the reader to get in touch with the genetic data.

Discussion

Obviously, as I am suggesting some improved analysis, this may impact the validity of their current findings. Still, I will make some broad comments on the discussion.

Too lenghty but see my suggestions which will help remove some of your more hypothetical and speculative discussion parts which lack data in support.

L. 382 “significant capacity” I think this may well be a significant overstatement of the current results? All in all, the impact of your drove road is relatively limited to Fis. The position of settlements in the landscape has a larger impact on genetic diversity. I would angle the discussion as such, framing it as in the reworked introduction, balancing functional and structural connectivity.

L. 337-344 This is too speculative. You have no data of your own in this manuscript which could support this. Also, would this high level of pollen flow in the drove, if this is indeed an explanatory mechanism, then not imply clear, higher levels of genetic diversity in the drove? Which was obviously not the case (though Fig. 3 may indeed suggest some non-significant trends in genetic diversity. Bear in mind however that you only have 6 and 7 populations in the drove/landscape, so results should be interpreted with care, and I would not put much faith in non-significant trends).

L. 360-375 As above, I would omit this because of the same argument: I would not put much faith in non-significant trends when only dealing with 13 populations in total. Stick to what you do have.

L. 404-427 There is redundancy in these paragraphs, and they could easily be merged and shortened.

Additional comments

Dear authors,

I have read this manuscript with great interest, as I have submerged myself over the past few years in the importance of domestic cattle as seed dispersal vectors, positively (?) impacting both species and genetic diversity in isolated plant communties. I believe that the paper could make a nice contribution to add more studies underlining the importance of domestic livestock as key providers of functional connectivity in fragmented landscapes. Nevertheless, some work remains to clarify the manuscript further to reach a level acceptable for eventual publication (see my major comments).

Besides my my comments in the remainder of my review, I list a series of minor comments here:

L. 82-3 “This will … on human well-being”. This sentence skips a few steps as to how biodiversity is relevant to human well-being. Elaborate or omit.
L. 83-86 Omit.
L. 105-107 “…, hence … Shannon et al. 2009).” Omit.
L. 115 “fragmentation-derived viability” Rephrase. No clue what the authors mean.
L. 127-132 The system in Germany may not be a pure transhumance system such as the one in the paper, but its sheperding system is still more a transhumance system where animals physically move through the landscapes, compared to true rotational grazing systems in other countries, such as in the Viroin in Belgium (Honnay et al. 2006) or the Stockholm archipelago in Sweden (Auffret et al. 2012, Plue & Cousins 2018), where animals are moved in trucks or boats between their pastures. So, I would not put it as sharply that your transhumance and Rico’s sheperding systems are so different.
L. 133 ?? Omit, but keep the part on L. 134, thát is relevant.
L 137 “were strong enough to” Omit.
L. 139 It cannot be that difficult to give those unpublished data, or a summary, to show how many seeds of Plantago are being dispersed. If anything, this may be important to learn how much seeds could be dispersed, putting an important nuance on the strength of the genetic imprint.
L. 147 “genetic differentiation” was not properly introduced, so that this is the first RQ comes as a total surprise.
L. 156 Is Plantago a diploid? Is it gynodioecous like other Plantago species, what would that mean?
L. 158 The species has a soil seed bank. What does that imply for the vegetation genetic patterns? A soil seed bank may slow down genetic changes to the aboveground vegetation (Honnay et al. 2008, Oikos or Plue et al. 2017, Annals of Botany). I think this could be inserted in the discussion, particularly in explaining why you see differences in the Fis but not in genetic diversity?
L. 166 “faster dynamics in annual plants” You mean rapid turn-over in generations?
L. 177 The drove road. A picture would really help, as I have no clue to imagine what such a “road” constitutes. Based on Google, it looks more like a long, quite broad grassland strip crossing the landscape. Most people, have no clue about what your system looks like.
L. 193 Not 20-30, 8-30 to be precise.
L. 199 As the drove road is a road, I wonder how populations where defined? How to limit where to sample?
Table 6, Fig.3 Can be merged to represent the most key information.
Fig. 1 I would like to see a topographical map underlaying the datapoints. Gives an idea of landscape heterogeneity.

I wish you good luck with the revisions,
dr. Jan Plue

·

Basic reporting

Although I am not a native English speaker, I find the manuscript very well-written with a clear and professional English language. The introduction is well structured, although I provided some advice to improve this structure a little bit more. The literature is relevant and updated. Structure conforms to PeerJ standards. Raw data are supplied. Figures require some qualitative improvement.

Experimental design

The sampling design is appropriate with a reasonable number of populations and enough individuals sampled in each population. The main hypothesis is clearly identified in the introduction (Lines 130-132) and the authors did a great job highlighting the knowledge gap that this study fills. I have some concerns with the statistical analyses, particularly with the potential effects that null alleles could have on the estimates of inbreeding and genetic diversity. I also miss a formal analysis relating the genetic differentiation of populations with landscape attributes (presence of drove roads) using landscape genetic tools.

Validity of the findings

Again, I think that new estimates accounting for the potential presence of null alleles in the markers used could provide more reliable genetic estimates. I also think that some speculation could be better phrased to be better identified as speculation.

Additional comments

Garcia-Fernandez and collaborators present an interesting work that investigates an original question. Specifically, they examined whether drove roads are facilitating livestock-mediated seed dispersal using a population genetic approach. The manuscript is well-written, and the results have important implications for ecosystem management. Therefore, I think that this work should be interesting for a broad audience ranging from ecologists to policy-makers. However, I have some concerns with the statistical analyses used that could influence the validity of their results (see METHODS section). I tried to provide some advice in this regard.

Comments for the authors

ABSTRACT
Major comment
Lines 38-43: I found this part of the abstract too speculative. The authors provide information on the genetic diversity and genetic differentiation of populations, but they did not use either indirect or direct methods to investigate gene flow (e.g. paternity or maternity analysis, historic gene dispersal). Of course, gene flow among populations could be a critical factor influencing their patterns of genetic differentiation and genetic diversity, but there are other factors that could be also impacting (e.g. population sizes, historical processes of colonization). I think that it is fine to interpret your results in terms of differences in gene flow, but I don’t think that you can distinguish between pollen-mediated gene flow and seed dispersal as you state in the lines 38-41.

Minor comments
Line 33: I would replace “genetic descriptors” with “genetic diversity and inbreeding coefficients” to be more specific.
Line 43: Given that the main topic is livestock-mediated dispersal and the term “human dispersal” has not been previously defined, it is not straightforward to understand what the authors mean with human dispersal, at least for a reader not specialized on this topic.

INTRODUCTION
The introduction is well-written, developing properly the main topics relevant for the scope of this work. Furthermore, the literature is well referenced, updated, and relevant for this work.

Minor comment
I would suggest improving a little bit the structure reducing the number of paragraphs. Specifically, I think that the first and second paragraphs are very similar and thus, they could be combined into a single one focused on habitat fragmentation as an important threat to genetic biodiversity.
In the same way, I would combine the third and fourth paragraphs into one moving from dispersal as general concept to the more specific mode of dispersal mediated by livestock. Doing this, I think that you quickly introduce the reader on the main topic of the manuscript.

METHODS

Major comment 1
I think that the authors could take advantage of their INEST analysis to obtain more accurate information on genetic diversity and inbreeding. First, some studies have reported better performance of INEST than other software (e.g. Micro-checker; Campagne et al. 2012 in MolEcolRes) estimating null allele frequencies. Second, INEST provides estimates of genetic diversity (He and Ho) and inbreeding coefficients accounting for null allele frequencies in the markers. Third, by comparing a model incorporating and excluding inbreeding as a covariate, you can explicitly test if inbreeding is significant in your populations. I truly think that the manuscript would be beneficiated from this valuable information.

Major comment 2
I have the following concerns with the STRUCTURE analysis:
1. Given the low genetic differentiation among P. lagopus populations, I would suggest including geographic sampling locations as prior population information in our STRUCTURE analysis, as it is recommended in Hubisz et al. 2009.
2. I found quite modest the number of permutations during the burn-in and MCMC (105 and 106). Maybe is a typo and authors meant 10 5 and 10 6; in that case, they should clarify it.
3. Given the low genetic differentiation of the populations, I wonder if the Evanno’s K selection criteria is the most appropriated. As can be observed in the individual bar plots (Fig 2 B and C), admixture is huge. The main disadvantage of Evanno’s method is that you never examine the possibility of K=1 (panmixia). One option could be to examine the mean Ln P(K) and select that K at which the curve reaches the plateau. However, there are other indexes that outperform both delta K or mean Ln P(K) with unevenly sampled collection sites (Puechmaille 2016; Li and Liu 2018 for software implementation of the indexes described in the previous work). For that reason, increasingly more works are encouraging the use of several K selection methods in order to get a more accurate “true” K.

Major comment 3
Given that the authors mentioned the use of a landscape genetics approach (line 145), I miss a formal test examining whether the genetic differentiation among populations is related to the continuous variation across the landscape regarding drove roads. In other words, using a landcover map, you can calculate conductance distances among your populations assuming that drove roads act as corridors facilitating gene flow (i.e., conductance distance). Then you can run an analysis (e.g. Mantel test, MLPE, causal modeling) relating the population pairwise genetic differentiation matrix and the conductance distance matrix. In this way, you can examine straightforwardly your hypothesis of drove roads enhancing gene flow between populations due to livestock-mediated seed dispersal.
Other option, if you prefer to keep a categorical design is to estimate migration rates among your populations using software such as MIGRATE-N. In fact, you could compare migration rates estimated using MIGRATE-N and BAYESASS to compare historical versus contemporary dispersal, if you want to examine whether resident livestock is influencing contemporary genetic differentiation (Lines 186-188).

Minor comments
Line 177: Please, move “hereafter CDR” to line 140 where you define the term for the first time.
Lines 180-181: Does the study species disperse its seeds in June or November? With this information, maybe you could examine asymmetric dispersal in your populations. In other words, if the species disperse its seeds in June, I would expect that southern populations act as source populations and northern ones as sink populations.
Line 195: How many individuals were deposited? “Some individuals” sounds a little bit vague.
Lines 241-244: Why don’t you run separate Mantel test for populations in agriculture matrix and drove roads? Then you could examine if there is isolation by distance in each group of populations and what relationship has steeper slope (i.e., more restricted gene flow). Also, you could examine the four scenarios proposed by Hutchinson and Templeton (1999) regarding the relationship between genetic differentiation and geographic distance.
Line 267: Why don’t you include ploughing history as an explanatory variable in your model?

RESULTS
Minor comments
Line 274: Please, replace “small” with “low”.

DISCUSSION
In general, the discussion is well written, nicely-structured and with a clear take-home messages. In addition, I would like to highlight the good work that the authors have done regarding the implications for the conservation of ecosystem services. I would only suggest considering other factors such as population sizes, colonization history when interpreting their findings (Lines 337-341). Pollinator availability could be an important factor influencing inbreeding coefficients, but also the availability of co-flowering non-relatives (i.e., density of reproductive individuals, asynchrony in flowering phenologies of reproductive individuals).

Minor comments
Line 350-351: this sentence is not clear, please rephrase it.
Lines 369-370: “[…] reduce the occurrence of private alleles”. Considering the modest geographic extent of the study area and the number of markers used (i.e., six), I cannot agree with this statement. There are populations with more than 5 private alleles (LCR01, LMA08 and LMA12).
Lines 429-437: I find this paragraph a little bit out of place here. I would move it to the previous section because it is part of the discussion of your results.

CONCLUSIONS
Conclusions are well-stated and linked to the original research question. However, I think that talking of long-distance plant migration is not limited to the supporting results. Long-distance dispersal could be a mechanism reducing genetic differentiation among your populations, but in my opinion, this hypothesis requires a different analytical approach involving direct estimates of dispersal distances.

FIGURES

Figure 1: I would use a landcover map to explicitly show the location of the drove roads and land use types in the surrounding matrix.

Figure 2: The resolution of this figure should be improved, it is blurred.

Figure 3: It would be nice to increase font size in the axis titles and tick labels. Also, I would replace “I” with “Shannon Index” in the plot “c”.

TABLES
Table 5: Please, include a column stating whether the populations is in a Drove Road or into the agricultural matrix.

Table 6: I think that the authors meant “Genetic descriptors” instead of “Inbreeding coefficient” in the first cell of the table.

REFERENCES
Campagne, P., Smouse, P. E., Varouchas, G., Silvain, J. F., & Leru, B. (2012). Comparing the van Oosterhout and Chybicki‐Burczyk methods of estimating null allele frequencies for inbred populations. Molecular Ecology Resources, 12(6), 975-982.
Hubisz, M. J., Falush, D., Stephens, M., & Pritchard, J. K. (2009). Inferring weak population structure with the assistance of sample group information. Molecular ecology resources, 9(5), 1322-1332.
Hutchison, D. W., & Templeton, A. R. (1999). Correlation of pairwise genetic and geographic distance measures: inferring the relative influences of gene flow and drift on the distribution of genetic variability. Evolution, 53(6), 1898-1914.
Li YL, Liu JX (2018) StructureSelector: A web-based software to select and visualize the optimal number of clusters using multiple methods.Molecular Ecology Resources, 18:176–177.
Puechmaille, S. J. (2016). The program structure does not reliably recover the correct population structure when sampling is uneven: subsampling and new estimators alleviate the problem. Molecular Ecology Resources, 16(3), 608-627.

RECOMMENDED SOFTWARE FOR ESTIMATION OF MIGRATION RATES AMONG POPULATIONS

MIGRATE-N: https://popgen.sc.fsu.edu/Migrate/Migrate-n.html
BAYESASS: http://www.rannala.org/software/

---

## Round 0.2 · Minor Revisions

Both reviewers are enthusiastic about your work and believe that the revised manuscript has been substantially improved. Please address their remaining suggestions and clarify how you assessed linkage disequilibrium between loci (see my previous decision letter).

·

Basic reporting

The submission is self-contained including all results relevant to the hypothesis that the authors examine. The authors did a great job improving the structure of the introduction and discussion substantially improving the readability of the manuscript. The literature is relevant and updated. The structure conforms to PeerJ standards. Raw data are supplied. Figures still require some qualitative improvement.

Experimental design

As I stated in my previous revision, I think that the sampling is appropriated and the mean hypotheses clearly identified. In the current version of the manuscript, the authors did a good job incorporating new analyses that provide estimates of inbreeding after accounting for null alleles. Although they did not explicitly examine the possibility of isolation by resistance, they did an important effort to include landscape features in their analyses.

Validity of the findings

Although I still think that there is some speculation regarding the effects of drove roads on pollination, the results are strong enough and of interest of a broad audience involving ecologists, policy-makers and managers.

Additional comments

Dear authors,
I really like the outcome of your revision. I think that both introduction and discussion have been substantially improved with a clear structure and a nice flow of ideas. Although I still think that isolation by resistance test would be appropriate, I see the merit of the authors incorporating landscape features in their analyses. Finally, I think that consideration of null allele frequencies in their markers results in more reliable estimates of inbreeding. Therefore, I only have minor comments aimed at providing some feedback to improve the manuscript.

Abstract
L28: What do you mean by “productivity”? Generation of biomass in an ecosystem?
L42: Without any quantitative comparison of pollinator communities between drove roads and agricultural matrix, I still find too speculative this statement.
Introduction
L52 “This genetic impoverishment can limit evolutionary potential.”
L64: What do you mean by “subjected to strong adaptation pressures”?
L68-69: I do not know if I agree with this statement. For example, overpopulation of ungulates in many regions has been reported by numerous studies. I think that this sentence should be rephrased to focus specifically on those groups exhibiting population declines (Large frugivores?)
L76-79: This sentence is quite wordy, try to synthesize it.
L124-134: I think that this paragraph is not necessary. You could include two sentences in the previous paragraph, one indicating the effects of drove roads in pollination and another explaining the effects of persistent seed banks.

Methods
L248: Please, indicate the number of microsatellite markers used in your analyses.
L260: Did you compare the models including and excluding inbreeding? This test is very useful to examine if there is evidence of inbreeding in each of your populations. You should get this information in the output of the analysis.

Results
L373-375: It would be good to provide at least a quantitative estimate such as the coefficient of variation.
A few comments about the results from Bayesass and Migrate:
You should have a migration rare of each population with the remaining populations, but I only see a migration rate for each population. Did you estimate the mean migration rate of each population with the remaining ones? If so, you should indicate it. One option to make more visual the information included in Table 6 could be a network plot. You could be interested in the approach used in Castilla et al. (2016) where migration rates are presented in this way. Sorry for the self-advertisement and of course no need of citing this work!
You must provide the standard errors of your estimates to check if they overlap zero and thus migration is negligible.
It is interesting the difference in the magnitude of historical and recent migration rates. It seems like migration was intenser in the past according to your estimates. I wonder if the decline in the use of drove roads could be linked with this reduction in the recent migration. How do the authors interpret this result?

Discussion
L546-549: Maybe an idea for these future studies could be to estimate mean dispersal distance separately for populations in drove roads and agricultural matrix using the approach implemented in the software SPAGeDi. Probably you will need to increase sampling size (sampling area and number of plants).

L550-552: I do not catch the message in this sentence. Please, rephrase it specifying the ecosystem functions provided by mobile pastoralism and what covers more than a half of the global land.

Figures and Tables
The number of tables is too high. I provide some suggestions in this regard:
1. Move Table 2 to supplementary material.
2. Add geographic coordinates of populations to table 3 and remove table 1. The rest of the information in Table 1 is redundant.
3. Move Table 5 to supplementary material.
4. I think that you could include the information of tables 7, 8 and 9 into a single table. The structure of the models is practically identical. You can maintain the explanatory variables in the first column, then two columns for each response variable including its corresponding coefficient and p-value.

I think that the figures still require some improvement.
Although Figure 1 is illustrative of the landscape and human management in the study area, I don’t think that it is informative. One option could be to move it to supplementary material. Other option could be to combine Figure 1 and 2 into a single figure. I would suggest removing the second panel and maintain the panel with the Iberian Peninsula with the framed area and the landscape map showing the spatial location of populations. Then I would add a third panel with the picture used in Figure 1.
Figure 4: I would suggest removing the grid lines inside the panels and increase font size.

References
Castilla, A. R., Pope, N., Jaffé, R., & Jha, S. (2016). Elevation, not deforestation, promotes genetic differentiation in a pioneer tropical tree. PloS one, 11(6), e0156694.

Reviewer 3 ·

Basic reporting

Please check the grammar of some sentences, I found some sentences that miss or have an extra word. Also some sentences lack clarity or I repeated.

Specifically:

Ln55: add: and microevolutionary...
Ln58: add by: is not always possible by just....
Ln60: instead of using often limiting replace by: are needed to sustain
Ln67: remove “measures”
Ln74: I will add to these references, one more recent.
Ln81: add, such as...
Ln86: I am not sure if the word Traditional is the correct here, would it be more: Historic routes
Ln108: rotational
Ln124-127: I don’t understand what meant by “rather than with the use of the landscape that seed dispersal depends on”. Pollinators also will be affected by landscape use, or does the authors meant specifically by the livestock herded associated dispersal? Please clarify as also I don’t get the idea of the following sentence.
Lines 219 and 276 with 278 and 292 have repeated ideas, foe e.g. calculation of FST with FreeNA and the calculation of Mantel test, please revise and correct.

Experimental design

I have read the revised version of this manuscript and the comments provided by two reviewers. This is the first time I have revised this manuscript and I found that the authors have made a good effort to address each of the comments, the analyses added made the work statistically more sound, so I don´t have further suggestions for analyses.

My only comment on this aspect is regarding the justification of the estimation of contemporary and historical migration rates. After reading the response letter it is clear the purpose and the new information these methods will add to the manuscript, but I don’t think the authors make enough clear within the text, the specific purpose to perform them.

Other comments in methods:
Ln340: Software or R packages to run these analyses?
Ln393: That’s a perfect linear fit! r=1, is that right?

Validity of the findings

Results are sound and the conclusions are well supported based on their own data and available studies.

Additional comments

This is an interesting work adding more evidence on the importance of livestock mediated seed dispersal on gene flow and genetic diversity in agricultural landscapes. As I stated above, after carefully reading the revised version I found the manuscript improved, the methods are sound and the discussion is in line with the provided evidence. I just have few comments that could improve the quality of this work, and the main observation I would make relates to the argument in lines (326-329), where the authors stress that allelic richness is a descriptor that was considered related to seed dispersal in similar studies of Rico et al 2014a, Rico & Wagner 2016, and DiLeo et al 2017. I don’t think this interpretation is that correct, in such studies allelic richness was used as the best estimate of genetic diversity to evaluate contemporary landscape effects compared to others, such as He., and not necessarily as a proxy linked or related to seed dispersal since also gene flow by pollen or population size are factors involved in genetic diversity estimates such as Ar. For FIS, in case of plants, it does have a pollen flow component which can be due to selfing or biparental inbreeding as specified in Rico et al. 2014. Mol Ecol, and in this work. Thus, I would omit the argumentation of the relation between RiqRare and seed dispersal,which would be incorrect, but certainly its valid to argue the use of this estimate (as did in ln312-314) to (at some level) contrast with previous studies.

Besides this comment I think this work merit publication as it contributes within its field.

---

## Round 0.3 · accepted · Accept

I am happy to accept your manuscript since you have successfully addressed all the issues raised in the last round of reviews. I believe it is an important contribution highlighting the importance of herbivore corridors and mobile pastoralism for reducing inbreeding levels and facilitating long-distance dispersal in plants.

Although English writing in the main text seems appropriate, I found this was not the case for the abstract. I thus ask you to revise your abstract before publication (perhaps asking a native English speaker for help).